# Integration of Gut Microbiota with Transcriptomic and Metabolomic Profiling Reveals Growth Differences in Male Giant River Prawns (*Macrobrachium rosenbergii*)

**DOI:** 10.3390/ani14172539

**Published:** 2024-08-31

**Authors:** Quanxin Gao, Hao Huang, Peimin Liu, Xiuxin Zhao, Qiongying Tang, Zhenglong Xia, Miuying Cai, Rui Wang, Guanghua Huang, Shaokui Yi

**Affiliations:** 1College of Life Science, Huzhou University, Huzhou 313000, China; gaoqx2008@163.com (Q.G.); huanghao2638@126.com (H.H.); 17337852963@163.com (P.L.); weilai20220925@163.com (X.Z.); tangqy@zjhu.edu.cn (Q.T.); 2Jiangsu Shufeng Prawn Breeding Co., Ltd., Gaoyou 225654, China; zjhill@126.com (Z.X.); caimiaoying2024@163.com (M.C.); 3Guangxi Key Laboratory of Aquatic Genetic Breeding and Healthy Aquaculture, Guangxi Academy of Fishery Sciences, Nanning 530021, China; raywongxx@163.com

**Keywords:** *Macrobrachium rosenbergii*, growth, gut microbes, plasma metabolites, multiomics

## Abstract

**Simple Summary:**

The giant freshwater prawn (GFP; *Macrobrachium rosenbergii*) is a tropical aquatic animal with high economic value. However, we found that GFP in the same environment will vary in individuals, growth differences will affect the development of the industry. There are three types of body size in male GFP, namely, large, medium and small, which lead to differences in their phenotypes. Since the gut is responsible for nutrient digestion and the hemolymph is responsible for immunity, we believe that there is a close relationship between the gut, hemolymph, and phenotype. Therefore, we used different research methods to understand the causes of their body size differences. This study can provide a certain reference value for the GFP aquaculture industry and help to improve the development of GFP aquaculture industry.

**Abstract:**

The giant freshwater prawn (GFP; *Macrobrachium rosenbergii*), a tropical species cultured worldwide, has high market demand and economic value. Male GFP growth varies considerably; however, the mechanisms underlying these growth differences remain unclear. In this study, we collected gut and hemolymphatic samples of large (ML), medium (MM), and small (MS) male GFPs and used the 16S rRNA sequencing and liquid chromatography–mass spectrometry-based metabolomic methods to explore gut microbiota and metabolites associated with GFP growth. The dominant bacteria were Firmicutes and Proteobacteria; higher growth rates correlated with a higher Firmicutes/Bacteroides ratio. Serum metabolite levels significantly differed between the ML and MS groups. We also combined transcriptomics with integrative multiomic techniques to further elucidate systematic molecular mechanisms in the GFPs. The results revealed that *Faecalibacterium* and *Roseburia* may improve gut health in GFP through butyrate release, affecting physiological homeostasis and leading to metabolic variations related to GFP growth differences. Notably, our results provide novel, fundamental insights into the molecular networks connecting various genes, metabolites, microbes, and phenotypes in GFPs, facilitating the elucidation of differential growth mechanisms in GFPs.

## 1. Introduction

The giant freshwater prawn (GFP; *Macrobrachium rosenbergii*) is of great economic value due to several characteristics, such as tender meat, a rapid reproduction cycle, and fast growth [1,2,3]. Currently, GFPs are widely farmed worldwide, including in China, India, Thailand, and Vietnam [4]. GFPs demonstrate significant sexual size dimorphism, with males growing faster and larger than females under identical conditions. This study addresses the gap in understanding the underlying biological mechanisms of this differential growth, which is crucial for optimizing GFP farming practices. In particular, under identical breeding conditions, the average body mass of male GFPs is approximately twice that of female GFPs [5]. Furthermore, male GFP populations demonstrate large individual differences in body size, and body size plays a crucial role in the growth competition process [6]. The larger an individual, the more opportunities they have to obtain food resources. Moreover, older individuals considerably inhibit the growth of subordinate individuals. At present, differential growth patterns in male GFPs, termed “heterogeneous individual growth”, is a major bottleneck reducing the profitability of GFP farming; however, research on biological mechanisms underlying these patterns has been insufficient.

Many recent studies have demonstrated the significance of gut microbiota in the growth, development, and reproduction of animals; it regulates the digestion and absorption of intestinal nutrients and improves intestinal physiological functions [7]. In aquatic animals, gut bacterial composition is closely associated with their growth rate. Faster-growing individuals may have higher levels of intestinal probiotics [8,9,10,11]. Aquatic animals’ gut microbiota typically comprises aerobic bacteria, facultative anaerobic bacteria, and obligate anaerobic bacteria. These gut bacteria are present within the walls, mucosa, and contents of the intestines; moreover, the composition and structure of gut microflora largely depend on the host genetics, feeding habits, food sources, and living environment of aquatic animals [12,13,14]. Several studies have confirmed that during shrimp development, the symbiotic dynamic balance of intestinal microbiota depends on the development characteristics of the host such that the gut bacterial composition and the host facilitate each other’s growth [15,16,17]. In fast-growing *Penaeus monodon*, gut microflora exhibits high abundance of Gram-positive bacteria (mostly Firmicutes); in contrast, the intestinal bacterial structure of slow-growing *P. monodon* tends to be more complex and competitive [18]. In *P. vannamei*, gut microbial structure is significantly correlated with the growth and development; furthermore, gut microflora demonstrates structural changes during the breeding of high-yield *P. vannamei* [19]. As such, assessing interactions between GFP and its gut microflora structure is essential for promoting health and improving production performance in GFPs.

Metabolomics enables the comprehensive assessment of endogenous metabolites, along with systematic identification and quantification of metabolites [20,21]. Metabolomics has been widely used to study the complex metabolic mechanisms underlying endogenous or exogenous factors in aquatic animals [22,23,24]. As an energy storage substance similar to hemoglobin, hemolymph regulates osmotic pressure in shrimp [25]. Therefore, the hemolymphatic metabolism index is a crucial indicator of the metabolic level, nutritional status, and disease resistance of male GFPs; as such, it can be used as an indicator of GFPs’ health status [26,27,28]. Evaluating plasma metabolite levels in male GFPs with different specifications may deepen the current understanding of a shrimp’s individual heterogeneous growth.

GFPs have long, thin, and simple intestines (Figure 1a), and their intestinal metabolism is extremely active and fast. Gut microbiota is essential for GFP growth and development [29]. However, the mechanisms underlying the roles of the gut microbiome on individual growth differences in male GFPs have not been fully characterized. Therefore, to clarify the relationship between gut microflora and individual heterogeneity among male GFPs of different sizes, we used 16S rRNA gene sequencing and metabolomic techniques to analyze gut microbial compositions and functions and further characterize their metabolomic profiles. Finally, we combined prestored transcriptomic analysis data to construct the interaction network modes among host genes, phenotypic traits, metabolites, and gut bacteria, improving the current understanding of the mechanisms underlying growth differences in GFPs.

## 2. Materials and Methods

### 2.1. Sample Collection

GFP individuals with identical genetic backgrounds were obtained from Jiangsu Shufeng Prawn Breeding (Gaoyou City, China). The GFP selected in this study were full-sib offspring, from the same male parent and female parent. All of the individuals were cultured under the same conditions and fed the same diet. Male individuals were selected to measure their BWs; here, the males were identified based on the presence of a genital pore (Figure 1b). On the basis of significant individual growth variations, male GFPs with the largest, medium, and smallest six weights were classified into ML, MM, and MS groups, respectively (Figure 1c). In the ML, MM, and MS groups, the average BWs were 64.2 ± 1.48, 40.8 ± 0.90, and 20.3 ± 0.70 g, respectively. The guts were excised, frozen in liquid nitrogen, and transferred to a −80 °C refrigerator before 16S rRNA sequencing. Hemolymph was collected and centrifuged at 6000× *g* at 4 °C for 15 min. The upper plasma layer was collected and frozen at −80 °C for metabolomic analysis. The gonad tissue was collected and stored at −80 °C for transcriptomic analysis.

All GFP males were cultured under identical conditions, and one individual each from the LM, MM, and MS groups was selected for the measurement of the following 13 phenotypic traits (Figure 1d–f): TW, BL, CL, CW1, CW2, CD, PL, PW, AL, AW1, AW2, abdominal depth, and second leg weight.

### 2.2. Bacterial Genomic DNA Extraction and Sequencing

Six male GFPs of different sizes were selected randomly, and their total gut bacterial DNA was extracted using an E.Z.N.A. Soil DNA kit. DNA quality was detected through 10 g/L agarose gel electrophoresis, and DNA concentration and purity were determined on a NanoDrop 2000. The V3–V4 variable region of 16S rRNA was amplified using the primers 341F (5′-CCTACGGGNGGCWGCAG-3′) and 805R (5′-GACTACHVGGGTATCTAATCC-3′) through the polymerase chain reaction with an upstream primer (5′-ACTCCTACGGGAGGCAGCA-3′) and downstream primer (5′-GGACTACHVGGGTWTCTAAT-3′). The amplified products were subjected to paired-end sequencing (250 bp) on the Illumina NovaSeq 6000 platform at Shanghai Paisenuo Biotechnology (Shanghai, China).

### 2.3. Metabolite Extraction

All plasma samples were thawed at 4 °C and swirled for 1 min for even mixing. Next, an appropriate amount of plasma was transferred into a 2 mL centrifuge tube, followed by the addition of 400 μL of methanol (stored at −20 °C) and vortex mixing for 1 min. This mixture was then centrifuged at 12,000× *g* at 4 °C for 10 min, and the supernatant was transferred to a 2 mL tube. A 4 ppm 2-chloro-L-phenylalanine solution prepared with 80% methanol and water (stored at 4 °C) was added. The supernatant was then filtered using a 0.22 μm filter. Finally, the filtrate was collected for liquid chromatography–tandem mass spectrometry (LC-MS/MS)-based detection.

### 2.4. LC-MS/MS Analysis

LC-MS/MS analysis was performed on the ultrahigh-performance liquid phase system Thermo Vanquish (Thermo Fisher Scientific, Waltham, MA, USA), equipped with an ACQUITY UPLC HSST3 column (2.1 × 150 mm, 1.8 μm; Waters, Milford, MA, USA), Thermo Orbitrap Exploris 120 mass spectrometry detector (Thermo Fisher Scientific, USA), electrospray ion source (ESI), as well as positive and negative ion modes. The Proteowizard (version 3.0. 8789) package tool MSConvert was used to convert the original mass spectrometry file into the mzXML format. The software package XCMS (V3.12.0) was used for peak detection, peak filtering, and peak alignment, and a quantitative list of substances was obtained. The substances were identified using the Human Metabolome Database, Metlin, MassBank, Lipid Maps, mzCloud, and KEGG databases, as well as a self-built database; the parameters were set at a ppm of <30. We used the lose signal correction method based on quality control samples to perform data correction and eliminate systematic errors. Materials with a relative standard deviation of >30% in quality control samples were filtered out.

We performed principal component analysis, PLS-DA, and orthogonal PLS-DA (OPLS-DA) to reduce the dimensions of the sample data using the R software (V4.0.3) package. The values of variable importance in the projection were obtained using OPLS-DA. A heatmap was plotted using the R package pheatmap, and the metabolites were clustered bidirectionally. Next, pathway analysis was performed using MetaboAnalyst, a web-based tool for the visualization of metabolomic data. All enriched pathways were scanned for differential metabolites and pathway maps using the visualization tool KEGG Mapper.

### 2.5. Correlation Analysis

We assessed Pearson correlations among phenotypic traits, gut microflora, and plasma metabolites and transcripts. The data were visualized using a thermograph in the R package gplot. *p* < 0.05 was considered to indicate statistical significance. Finally, in network analysis, we used the estimateNetwork function to fit the network model.

## 3. Results

### 3.1. Gut Microbial Community Composition

A Venn diagram for the ML, MM, and MS groups (at the OTU level) is illustrated in Figure 2a. There were 720, 621, and 533 OTUs present in the ML, MM, and MS group, respectively. A total of 92 OTUs (41%) were shared by all groups. To compare gut microflora compositions among groups, we selected the top 20 most abundant phylum and genus to plot a heatmap (Figure 2b). At the phylum level, the abundance of Proteobacteria, Fusobacteria, and TM6 was the highest in the ML group; that of TM7, Gemmatimonadetes, Firmicutes, Tenericutes, Actinobacteria, Chlorobi, Nitrospirae, Fibrobacteres, OD1, and Chloroflexi was the highest in the MM group; and that of Deferribacteres, Bacteroidetes, and OP9 was the highest in the MS group. At the genus level, *Synechococcus*, *Aeromonas*, and *Rhodobacter* abundance was the highest in the ML group; as such, these gut bacteria may play a major role in GFP growth (Figure 2c).

In the large (ML), medium (MM), and small (MS) male GFP groups, the dominant bacterial phyla were Firmicutes (average abundance = 56.82%, 74.46%, and 62.42%, respectively) and Proteobacteria (average abundance = 41.52%, 22.14%, and 35.52%, respectively). The MM group demonstrated the highest and lowest abundance of Sclerenchyma and Proteobacteria, respectively. Actinobacteria abundance was higher in the MM group (2.58%) than in the ML (0.83%) and MS (1.50%) groups (Figure 2e). At the genus level, *Lactococcus* demonstrated the highest average abundance (Figure 2e), accounting for 55.16%, 67.58%, and 61.09% of all microorganisms in the ML, MM, and MS groups, respectively. In the MM group, the average abundance of *Lactococcus* (67.58%), *Enterococcus* (4.75%), and *Prauserella* (1.79%) was significantly higher than that in the other two groups. The Firmicutes/Bacteroides ratios in ML, MM, and MS groups were 0.56816/0.00058, 0.74463/0.0011, and 0.62424/0.00188 respectively, showing that ML (979.59) > MM (676.94) > MS (332.04). In general, the differences in intestinal microbiota structure were significant between male GFPs with different growth rates.

### 3.2. Identification and Analysis of Plasma Metabolites in Male GFP with Different Growth Rates

Partial least squares (PLS) regression was used to establish a relationship model between metabolite and sample categories for modeling prediction of sample category. To assess the model’s quality, we performed 200 response permutation tests on the PLS discriminant analysis (DA) model. As shown in Figure 3a,b, all three groups were clearly divided into three parts, confirming that the prediction model was reliable, and the repeatability for each sample in the same group differed significantly among the three groups.

To screen the key metabolites possibly mediating GFP growth performance, we used the top 20 abundant metabolites in each group to classify the functions of the metabolites. As presented in Table 1, L-tyrosine, phosphorylcholine, succinic acid, 2-ketobutyric acid, and acetylcholine demonstrated the highest abundance in the ML group; moreover, the abundance of plasma metabolites in the MM group was consistent with that in the ML group. Picolinic acid, maleic acid, O-phosphoethanolamine, L-norvaline, indican, and L-methionine demonstrated the highest abundance in the MS group.

The relative contents of the metabolites were measured based on the Z-score (Figure 3c). The results demonstrated that the relative contents of indoleglycerol phosphate, 4-phenylbutyrate, maleic acid, and citrulline were the highest in the MS group, whereas the relative contents of other metabolites were relatively consistent among all groups. We further performed hierarchical cluster analysis with the relative values of metabolites as metabolic levels to analyze different metabolites in GFPs with different growth rates. As illustrated in Figure 3d, the metabolites among the three groups demonstrated varied metabolic patterns and metabolite levels. Notably, the levels of most metabolites in the ML and MS groups demonstrated contrasting trends. The levels of L-ribulose, indican, aminocaproic acid, L-methionine, procollagen 5-hydroxy-L-lysine, and other metabolites were the lowest in the ML group; an opposite trend was noted in the MS group.

To explore the function of differential metabolites in GFP, we performed the Kyoto Encyclopedia of Genes and Genomes (KEGG) functional enrichment analysis and screened the top 20 pathway entries with significant enrichment for mapping. As shown in Figure 3e, protein digestion and absorption, biosynthesis of amino acids, tyrosine metabolism, aminoacyl-tRNA biosynthesis, biosynthesis of alkaloids derived from histidine and purine, glycerophospholipid metabolism, cAMP pathway, and regulation of actin cytoskeleton were enriched pathways associated with GFP growth performance. In our enrichment analysis, pathways related to amino acid metabolism and biosynthesis were detected widely. In addition, metabolites enriched in the metabolic pathways of protein digestion and absorption, amino acid biosynthesis, and mineral absorption demonstrated high levels in ML GFPs. Therefore, the metabolites associated with amino acid metabolism and biosynthesis may play a key role in male GFP growth performance.

### 3.3. Correlation between Gut Bacteria, Transcripts, Phenotypes, and Metabolism in Male GFPs

The correlations between phenotypes and metabolism were visualized using correlation heatmaps (Figure 4a). The results demonstrated that 20 of the top 53 abundant metabolites were positively correlated with GFP phenotypes. The metabolites 3-hydroxymethylglutaric acid, ethylmethylacetic acid, and imidazol-5-yl-pyruvate were highly and positively associated with the most phenotypes. Propylthiouracil, prostaglandin F3a, 8-shogaol, L-erythrulose, and L-ribulose were highly and negatively associated with phenotypes such as total weight (TW), palm length (PL), and palm width (PW). In terms of the correlation between gut bacteria and metabolism (Figure 4b), Candidatus Xiphinematobacter was found to be correlated highly and significantly positively with L-phenylalanine but correlated highly and negatively with 10-deoxygerfelin. Roseomonas demonstrated a high and negative correlation with -iditol, 5-(methylthio)-2,3-dioxopentyl phosphate, and 10-deoxygerfelin.

As illustrated in Figure 4c, we a performed correlation analysis between the top 30 microorganisms at the genus level and the phenotypes in male GFPs. *Faecalibacterium* was significantly and positively correlated with body length (BL), abdominal length (AL), body weight (BW), carapace length (CL), carapace depth (CD), carapace weight (CW1), carapace width (CW2), abdominal weight (AW1), abdominal width (AW2), and TW. Moreover, *Rubrobacter* was significantly and positively correlated with CL, CD, CW2, PW, and TW. These results indicated that *Faecalibacterium* and *Rubrobacter* may be crucial for phenotypic differentiation in GFPs. Regarding the correlation between gut bacteria and the 30 top differentially expressed genes (DEGs; Figure 4d), *Enterococcus* and *Blautia* were positively and negatively correlated with most DEGs, respectively. As such, these bacteria may affect growth by regulating gene expression.

### 3.4. Integrative Multiomic Analysis of Transcripts, Microbiome, Phenotypes, and Metabolites in Male GFPs

In the integrative multiomic analysis on the DEGs, the analytical results of phenotypes and plasma metabolomics provided two correlation models: DN1168_c1_g1–CW2–Rubrobacter and DN1168_c1_g1–CL–Faecalibacterium. Both models contributed to the apparent visualization of different omics (Figure 4e). The obtained association network demonstrated that the gene DN1168_c1_g1 is involved in most associations with phenotypes. To explain the differences in gut microbiota and plasma metabolites between GFPs with different growth rates, we integrated the correlative 16S rRNA gene sequencing results, metabolomic data, and phenotypes of male GFPs (Figure 4f). The results demonstrated that 20 metabolites and 11 microorganisms were involved in this correlation network, which also revealed 13 models of correlation between metabolites, microbiome, and phenotypes in male GFPs.

## 4. Discussion

Crustaceans typically demonstrate a large number of microbial communities in their guts. Gut microbiota has formed an interactive relationship with the host over the long course of evolution. As such, these microorganisms have become an indispensable part of their hosts. Gut microbiota structure is closely related to host health and growth performance, and intestinal microbiota structure is affected by many factors, such as nutritional substances, at different developmental stages and host genetics [30,31,32,33]. In the current study, we examined the differences in gut microbiota of male GFPs with different growth rates. Our 16S rRNA sequencing results demonstrated that gut microbiota in all groups was mainly of the phyla Proteobacteria and Firmicutes—consistent with previous results [34,35,36,37]. Gut Proteobacteria abundance is also associated with the performance of shrimp grown in an outdoor pond [38]. Many Firmicutes members can secrete active enzymes that facilitate the hydrolysis and utilization of carbohydrates and promote the absorption of energy by improving lipid metabolism [39,40]. Firmicutes and Bacteroides can affect fatty acid uptake and lipid metabolism in fish [41], and the Firmicutes/Bacteroides ratio is closely related to host energy metabolism, fat deposition, and growth. The higher the Firmicutes/Bacteroides ratio, the more efficient the host’s metabolism and growth [42]. The current results demonstrated that the Firmicutes/Bacteroides ratio was the highest in the ML group, followed by the MM group, and finally the MS group. As such, male GFPs with a higher growth rate may demonstrate better intestinal bacterial structure and higher nutritional metabolic efficiency.

At the genus level, *Lactococcus* accounted for the highest proportion of the gut bacterial structure in all GFP groups—similar to our previous results [43]. *Lactococcus* activates the innate immune system to produce cytokines and regulate the immune response [44,45,46]. Moreover, *Lactococcus* can regulate carbohydrate metabolism and absorption in livestock and poultry, as well as in mammals [47]. Therefore, in male GFPs, *Lactococcus* may affect growth and development by regulating carbohydrate metabolism, whereas the specific probiotic function of *Lactococcus* on growth warrants further exploration.

Metabolomics, the study of metabolic mechanisms, involves the measurement of metabolites in a whole biological sample and the elucidation of their potential metabolic pathways. Metabolites can directly fulfill the needs of individual growth and development through various metabolic reactions to synthesize energy and biological macromolecules; different metabolites and metabolic pathways affect the development of the related tissues and organs within a body [48,49,50]. In the current study, plasma metabolites in male GFPs of different sizes were investigated through LC-MS/MS. The results demonstrated that the abundance of plasma metabolites differed among GFPs of different sizes. In our hierarchical cluster analysis, we found that metabolite levels in large and small GFPs demonstrate opposite trends, indicating that metabolic patterns differ among GFPs with different growth rates.

Histidine contains an isopyrazole ring, the main component amino acid of functional proteins such as histone hemoglobin [24]. Small peptides composed of histidine and histamine have special physiological functions, and histidine plays a major role in metabolism and growth [51]. The levels of amino acid metabolites such as histidine and methionine were high in the ML group, and these metabolites were enriched in several metabolic pathways, such as protein digestion and absorption, amino acid biosynthesis, and mineral absorption. As such, histidine may be involved in various biochemical reactions within the body and thus be essential for GFP growth and development.

Our results further demonstrated that metabolites with significant differences were mainly involved in pathways such as alanine aspartate and glutamate metabolism, mineral absorption, protein digestion and absorption, and amino acid biosynthesis. Amino acid metabolism plays an important role in muscle cell growth promotion, improving the flavor of meats [52]. Moreover, we observed higher levels of many amino acids in fast-growing male GFPs. Therefore, we speculate that metabolites related to amino acid biosynthesis, alanine aspartate and glutamate metabolism, and protein absorption may play a major role in the growth performance of GFPs.

Gut microbiota and host phenotypes demonstrate a strong symbiotic relationship [53]. Gut microbiota interacts with various physiological functions in the host through its metabolites, which are essential for host growth and development [54]. In our integrated analysis of the microbiome and metabolome, *Candidatus Xiphinematobacter* was significantly correlated with six of the top thirty metabolites, which might be involved in metabolism regulation and, in turn, affect growth in GFPs. However, we also noted that *Faecalibacterium* and *Rubrobacter* were most significantly associated with GFP phenotypes. *Faecalibacterium* and *Roseburia* are major producers of butyrate in the intestines [55]. Gut microbiota such as Faecalibacterium and Roseburia may have enhanced growth by producing butyrate, which regulates host metabolism and immune responses. This modulation of metabolic pathways, particularly those involving amino acid biosynthesis and protein absorption, is crucial for the growth differences observed in GFPs.

A main strength of our study is that we considered all interactions between multiple biological levels and their associations. In the integrated analysis of the microbiome, metabolome, and transcriptome, we constructed two association networks. The results demonstrated that the gene DN1168_c1_g1 was involved in the most associations with phenotypes. Moreover, *Faecalibacterium* and *Roseburia* were strongly correlated with the aforementioned gene, as well as CW2 and CW1. As such, two correlation models, DN1168_c1_g1–CW2–*Rubrobacter* and DN1168_c1_g1–CL–*Faecalibacterium*, were constructed. In addition to *Faecalibacterium* and *Roseburia*, *Bacteroides* were associated with a wide variety of metabolites, and these metabolites were significantly correlated with various phenotypes. In aquatic animals, *Bacteroides* is a prominent bacterial group in gut microbiota, with roles related to symbiosis and pathogenicity [56]. Taken together, these results indicate that symbiotic bacteria affect male GFP growth and development by regulating host metabolism and gene expression.

## 5. Conclusions

In this study, we identified gut microbial taxa and metabolites associated with GFP phenotypes and used integrative multiomic analysis, combining the aforementioned results with transcriptomic data, to further explore the differential growth mechanisms among male GFPs. Our results indicate that the intestinal symbiotic bacteria can influence GFP growth by modulating metabolism and gene expression, as well as their correlations. This provides a novel paradigm to explore potential mechanisms underlying differential growth in GFPs. Moreover, we constructed an interaction network to gain insight into the crosstalk between genes, metabolites, and gut bacteria during GFP growth. The current results, at the integrated omics level, provide an understanding of the mechanisms underlying GFP growth.

## Figures and Tables

**Figure 1 animals-14-02539-f001:**
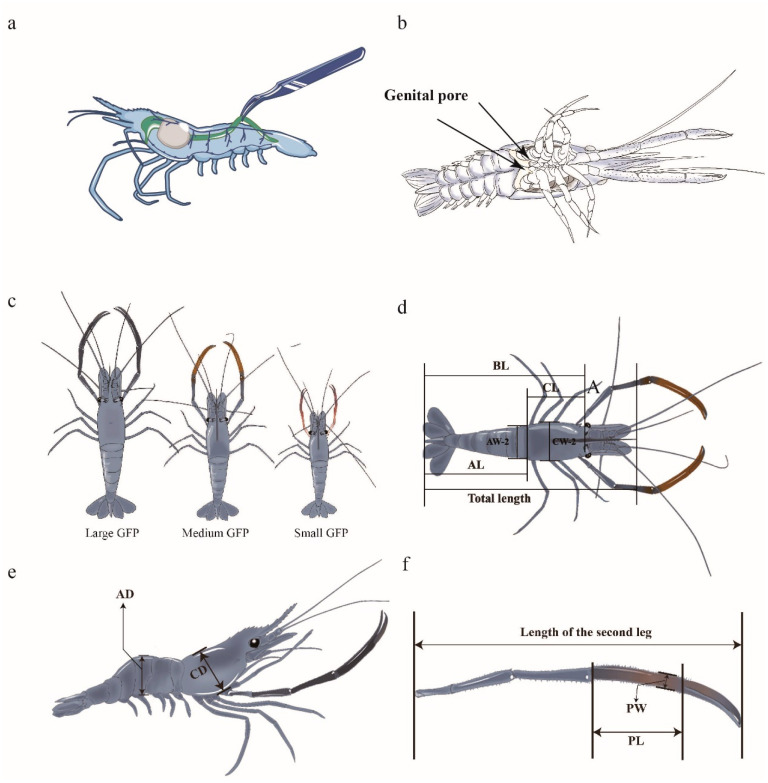
Phenotypic traits of GFPs used in this study. (**a**) Location of GFP gut. (**b**) Male genital pore (two arrows). (**c**) Male GFPs of different sizes. (**d**) Dorsal view of a GFP (BL: body length, CL: carapace length, AL: abdominal length, AW-2: abdominal width, CW-2: carapace width). (**e**) Lateral view of a GFP (AD: abdominal depth, CD: carapace depth). (**f**) Second leg of a GFP (PL: palm length; PW: palm width).

**Figure 2 animals-14-02539-f002:**
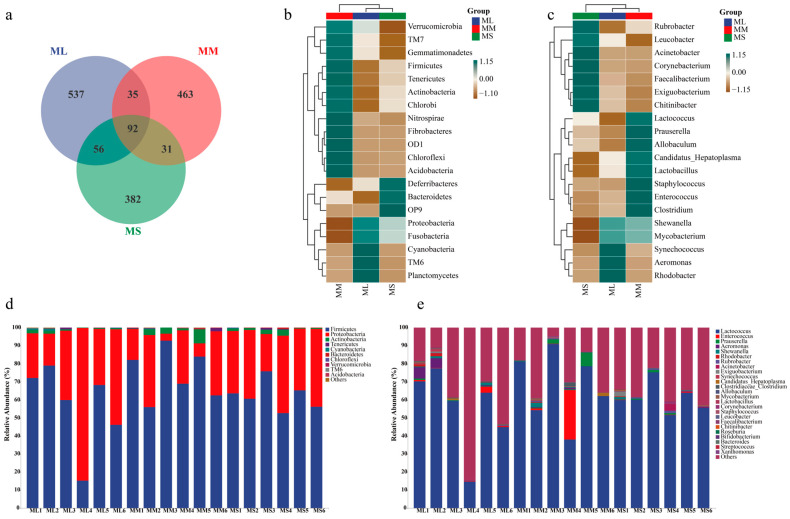
Relative abundance of gut microflora in male GFPs with different growth rates. (**a**) Venn diagram. (**b**) Abundance of gut microflora at phylum. (**c**) Abundance of gut microflora at genus levels. (**d**) Heatmap of the most abundant bacteria at phylum level. (**e**) Heatmap of the most abundant bacteria genera.

**Figure 3 animals-14-02539-f003:**
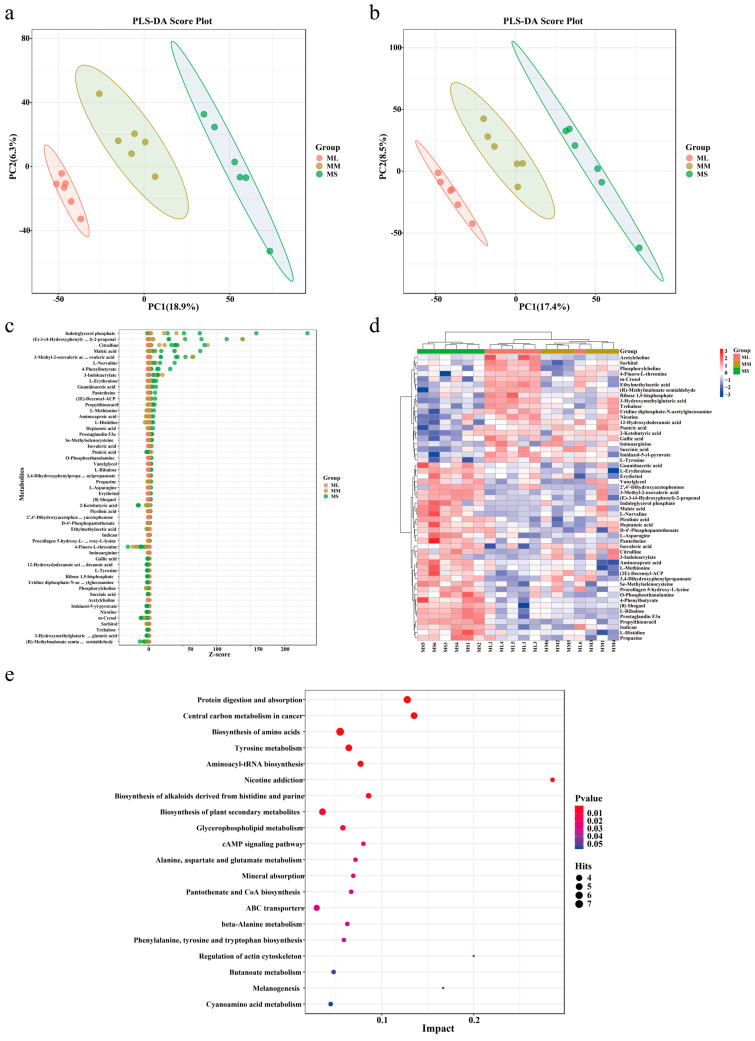
Identification and analysis of different metabolites and pathway enrichment of male GFPs with different growth rates. Score plots of our PLS-DA model for plasma metabolites in the (**a**) positive and (**b**) negative ion modes. Identification of different metabolites based on a Bubble map (**c**) and a Heatmap (**d**). (**e**) Pathway enrichment analysis of differential metabolites.

**Figure 4 animals-14-02539-f004:**
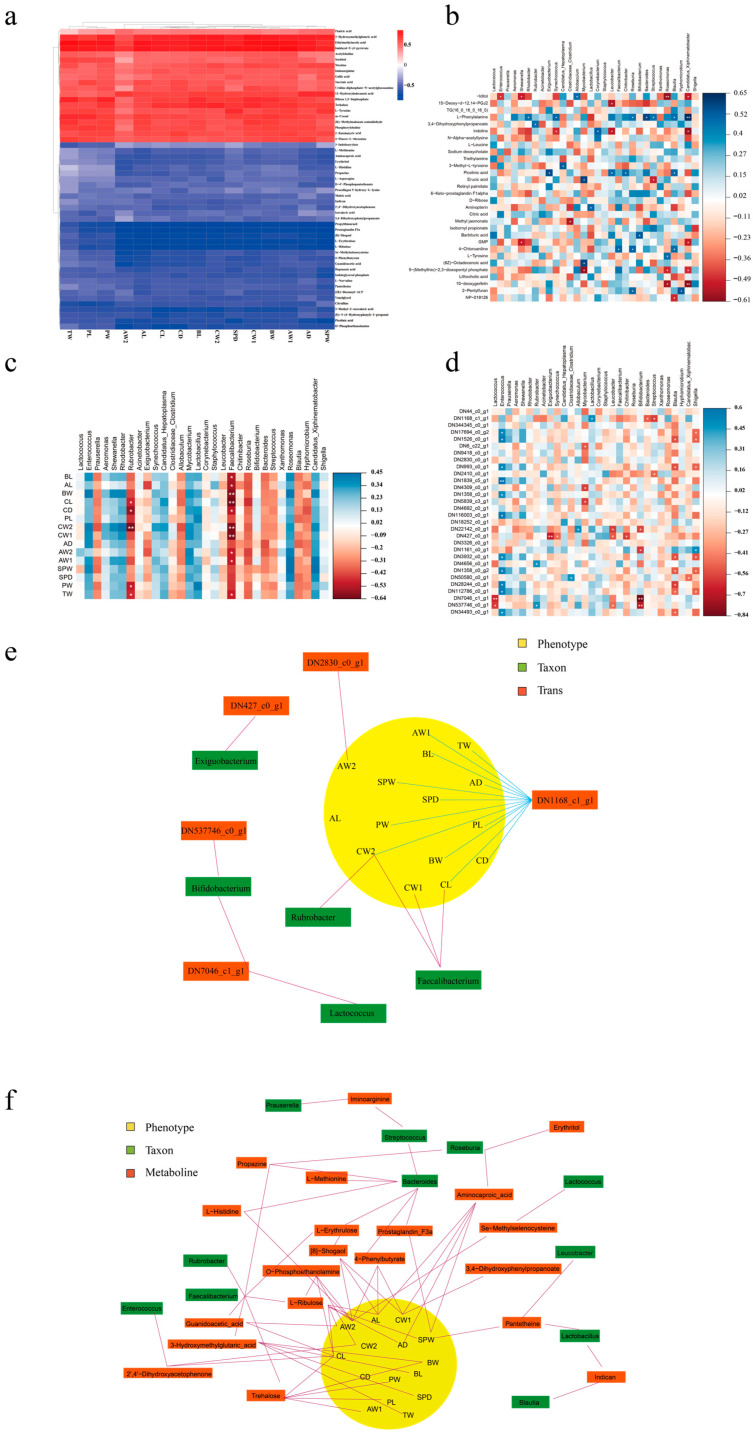
Integrative multiomic analysis. Heatmap correlation analysis between (**a**) phenotypes and metabolites, (**b**) microbiome and metabolites, (**c**) microbiome and phenotypes, and (**d**) microbiome and transcripts (* *p* < 0.05, ** *p* < 0.01). (**e**) Network associations among DEGs, phenotypes, and microbiome. (**f**) Network associations among metabolites, phenotypes, and microbiome.

**Table 1 animals-14-02539-t001:** Abundance of key plasma metabolites in GFPs.

Metabolite	Pathway	Average Abundance in Each Group		
		ML	MM	MS
Picolinic acid	Tryptophan metabolism	2,512,533,397	2,783,737,035	3,886,922,059
L-Tyrosine	Methane metabolism	1,380,214,938	818,298,529.6	494,550,907.6
Succinic acid	Biosynthesis of terpenoids and steroids	207,214,268.1	138,050,153	31,549,879.95
Aminocaproic acid	Caprolactam degradation	162,473,612.3	143,181,951.9	220,255,718.7
Phosphorylcholine	Glycerophospholipid metabolism	149,938,816	95,324,908.67	91,402,453.03
L-Methionine	Mineral absorption	126,393,703	116,471,294.1	183,369,889
L-Histidine	Aminoacyl-tRNA biosynthesis	65,508,706.11	63,207,424.53	68,971,672.67
L-Asparagine	Protein digestion and absorption	54,059,698.19	47,438,258.65	79,577,739.17
O-Phosphoethanolamine	Sphingolipid pathway	50,654,144.98	49,672,681.24	70,910,504.34
Acetylcholine	Taste transduction	39,034,131.15	19,257,807.04	21,312,743.56
Sorbitol	ABC transporters	24,872,146.19	17,401,326.46	19,175,258.42
L-Norvaline	\	20,880,672.75	40,609,717.36	110,409,043
Citrulline	Arginine biosynthesis	20,244,481.76	95,438,311.48	124,961,442.1
Ribose 1,5-bisphosphate	Phosphonate and phosphinate metabolism	11,435,523.25	9,671,995.33	8,929,543.9
2-Ketobutyric acid	Biosynthesis of amino acids	10,922,998.81	8,515,497.23	4,952,491.33
L-Ribulose	Pentose and glucuronate interconversions	3,659,200.9	3,401,882.38	7,809,899.52
Trehalose	Phosphotransferase system (PTS)	1,774,509.08	1614,878.04	1,183,823
Punicic acid	\	1,144,145.89	1,470,456.6	605,040.97
Maleic acid	Butanoate metabolism	1,045,720.79	5,604,645.41	16,113,880.62

## Data Availability

The data that support the findings of this study are available from the corresponding author upon reasonable request.

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
