# Peer review of "Integration of Gut Microbiota with Transcriptomic and Metabolomic Profiling Reveals Growth Differences in Male Giant River Prawns (Macrobrachium rosenbergii)"

_animals, 2024, doi:10.3390/ani14172539_

Round 1

Reviewer 1 Report

Comments and Suggestions for Authors

This study identified gut microbial taxa and metabolites associated with GFP phenotypes and used integrative multiomic analysis, combining the aforementioned results with transcriptomic data, to further explore the differential growth mechanisms among male GFPs. The current results, at the integrated omics level, provide a novel paradigm to explore potential mechanisms underlying differential growth in GFPs. It is an interesting study and can help us to understand the molecular mechanism underlying GFP growth, however, there are some issues that the authors still need to address before the paper is acceptable for publication. The detail information is as following: 

1.      In Abstract, Page: 1, Lines: 26-27.
Suggested Change:
Original: "The results demonstrated that in all three groups, the dominant bacteria were from the phyla Firmicutes and Proteobacteria; moreover, the higher the growth rate, the higher was the Firmicutes/Bacteroides ratio. Serum metabolite levels in the ML group significantly differed from those in the MS group."
Revised: "The dominant bacteria were Firmicutes and Proteobacteria; higher growth rates correlated with a higher Firmicutes/Bacteroides ratio. Serum metabolite levels significantly differed between the ML and MS groups."

2.      In Introduction, Page: 2, Lines: 44-45.
Suggested Change:
Original: "GFPs demonstrate sexual size dimorphism, whereby males grow faster and are significantly larger than females."
Revised: "GFPs demonstrated significant sexual size dimorphism, with males growing faster and larger than females under identical conditions. This study addressed the gap in understanding the underlying biological mechanisms of this differential growth, which is crucial for optimizing GFP farming practices."

3.      In Methods, experimental Animals. There was no information provided about the genetic background of the animal population.  Details about the culture environment of these animals were not included in the manuscript. Other important information about the experimental animals, such as their ages, was also missing. Factors like rearing conditions, feeds, and feeding regimes significantly affect the animals' gut flora, but these were not addressed.

4.      In Methods, Page: 4, Lines: 120-121.
Comment: Specify the primers used for 16S rRNA gene sequencing.
Suggested Change:
Original: "The V3-V4(a) variable region of 16S rRNA was amplified through polymerase chain reaction."
Revised: "The V3-V4 variable region of 16S rRNA was amplified using the primers 341F (5′-CCTACGGGNGGCWGCAG-3′) and 805R (5′-GACTACHVGGGTATCTAATCC-3′) through polymerase chain reaction."

5.      In Methods, transcriptomic analysis was also performed in this study. Please explain what tissues were used.

6.      Page: 17
Lines: 336-337
Suggested Change:
Original: "Gut flora structure affects growth."
Revised: "Gut microbiota such as Faecalibacterium and Roseburia may have enhanced growth by producing butyrate, which regulates host metabolism and immune responses (Faden, 2022). This modulation of metabolic pathways, particularly those involving amino acid biosynthesis and protein absorption, is crucial for the growth differences observed in GFPs."

7.      In References, ensure that all references were correctly formatted according to the journal's guidelines

Comments on the Quality of English Language

In general, the manuscript is clearly written.

Author Response

Thanks for the reviewer's suggestions. The reply is uploaded to the attachment.

Reviewer 2 Report

Comments and Suggestions for Authors

The paper “Integration of gut microbiota with transcriptomic and metabolomic profiling reveals growth differences in male giant river 3 prawns (Macrobrachium rosenbergii)” has novelty information in the addressed área. However, some minor corrections should be taken into account.

Reviewer’s corrections:

Abstract

Line 30. I recommend using the term "plasma" instead of "serum" throughout the document.

Introduction

Lines 66: Please, put Gram-positive.

Line 69. Please, delete “of P. vannamei”, because it is redundant.

Materials and methods

Line 110: Was any anticoagulant used?... During centrifugation, the hemocytes were lysed?....therefore, we have plasma and intracellular material?... Please, put "g" forces.

Line 123: What equipment was used?...miniSeq, miSeq?...what was the length of the reads, 150, 300 bp?

Line 129. Please, put "g" forces.

Line 159. Please, put “P” in italics.

Results

Line 163. I suggest that an alpha and beta diversity analysis be done.

Line 166. It is important to mention the number of non-shared OTUs.

Line 172,173. Please, put genera and species in italics in all document.

Lines 177,178. Please put “at phylum level”.

Lines 178,179. Delete “genus levels” and put “genera”.

Discussion

Line 293. Firmicutes/Bacteroides ratio...Where are the results of this analysis?... This information is very important because of its relationship with the metabolism and growth of organisms.

References

Line 389. Please, put species in italics.

The corrections are indicated in the manuscript.

Author Response

(The authors gave the same response as above.)
